# SntB triggers the antioxidant pathways to regulate development and aflatoxin biosynthesis in *Aspergillus flavus*

Dandan Wu[1†], Chi Yang[1,2†], Yanfang Yao[1], Dongmei Ma[3], Hong Lin[1], Ling Hao[1], Wenwen Xin[4], Kangfu Ye[1], Minghui Sun[3], Yule Hu[1], Yanling Yang[1]*, Zhenhong Zhuang[1]*

[1]Key Laboratory of Pathogenic Fungi and Mycotoxins of Fujian Province, Key Laboratory of Biopesticide and Chemical Biology of Education Ministry, Proteomic Research Center, and School of Life Sciences, Fujian Agriculture and Forestry University, Fuzhou, China; [2]Institute of Edible Mushroom, Fujian Academy of Agricultural Sciences, Fuzhou, China; [3]College of Animal Sciences, Fujian Agriculture and Forestry University, Fuzhou, China; [4]State Key Laboratory of Pathogen and Biosecurity, Institute of Microbiology and Epidemiology, Academy of Military Medical Sciences (AMMS), Beijing, China

*For correspondence:
13635287532@163.com (YY);
ZH_Zhuang@fafu.edu.cn (ZZ)

†These authors contributed equally to this work

Competing interest: The authors declare that no competing interests exist.

## eLife assessment

In this **useful** study, the authors investigate the regulatory mechanisms related to toxin production and pathogenicity in Aspergillus flavus. Their observations indicate that the SntB protein regulates morphogenesis, aflatoxin biosynthesis, and the oxidative stress response. The data supporting the conclusions are **compelling** and contribute significantly the advancing the understanding of SntB function.

**Abstract** The epigenetic reader SntB was identified as an important transcriptional regulator of growth, development, and secondary metabolite synthesis in *Aspergillus flavus*. However, the underlying molecular mechanism is still unclear. In this study, by gene deletion and complementation, we found SntB is essential for mycelia growth, conidial production, sclerotia formation, aflatoxin synthesis, and host colonization. Chromatin immunoprecipitation sequencing (ChIP-seq) and RNA sequencing (RNA-seq) analysis revealed that SntB played key roles in oxidative stress response of *A. flavus*, influencing related gene activity, especially *catC* encoding catalase. SntB regulated the expression activity of *catC* with or without oxidative stress, and was related to the expression level of the secretory lipase (G4B84_008359). The deletion of *catC* showed that CatC participated in the regulation of fungal morphogenesis, reactive oxygen species (ROS) level, and aflatoxin production, and that CatC significantly regulated fungal sensitive reaction and AFB1 yield under oxidative stress. Our study revealed the potential machinery that SntB regulated fungal morphogenesis, mycotoxin anabolism, and fungal virulence through the axle of from H3K36me3 modification to fungal virulence and mycotoxin biosynthesis. The results of this study shed light into the SntB-mediated transcript regulation pathways of fungal mycotoxin anabolism and virulence, which provided potential strategy to control the contamination of *A. flavus* and its aflatoxins.

## Introduction

*Aspergillus flavus* is one of the common asexual species, a saprophytic fungus and the second largest pathogenic fungus after *Aspergillus fumigatus*, widely distributed in soil, air, water, plants, and agricultural products in nature (*Hedayati et al., 2007*). Aflatoxins produced by *A. flavus* have strong toxicity, and are extremely harmful to human society. Animal and human health can be negatively affected by aflatoxins, which are carcinogenic, teratogenic, and mutagenic (*Guan et al., 2021*). Among aflatoxins, AFB1 is the most frequently occurring and the most toxic and carcinogenic, which is converted to AFB1-8 and 9-epoxide in the liver and formed adducts with the guanine base of DNA and thus results in acute and chronic diseases in both human and household animals (*Mishra and Das, 2003*). According to the Food and Agriculture Organization of the United Nations, 25% of the food crops in the world are contaminated with aflatoxins (*Vardon et al., 2003*), which are often detected in grains, nuts, and spices (*Riba et al., 2010*; *Plaz Torres et al., 2020*). It is urgent to control the contamination of *A. flavus* and its main mycotoxin, AFB1.

In recent decades, the biosynthetic pathway of aflatoxins was investigated in detail benefit from the sequence of *A. flavus* genome (*Cleveland et al., 2009*). This pathway consists of a complex set of enzymatic reactions (*Yabe and Nakajima, 2004*; *Yu, 2012*; *Roze et al., 2013*; *Ehrlich, 2009*). In general, these enzymes are encoded by clusters of genes, which are regulated by cluster-specific genes: *aflR* and *aflS* (*Chang, 2003*; *Price et al., 2006*). The initial stage of aflatoxins biosynthesis is catalyzed by polyketide synthase (PKSA) to form the polyketone backbone (*Xu and Luo, 2003*). The synthesis of aflatoxins is additionally influenced by environmental stimuli such as pH, light exposure, nutrient availability, and the response to oxidative stress, potentially leading to the alteration of gene expression related to toxin biosynthesis (*Yu and Keller, 2005*; *Georgianna and Payne, 2009*; *Montibus et al., 2015*). Besides the biosynthetic pathway and its internal gene regulation, protein post-translational modifications, an important mean of epigenetics, represent an important role in the regulation of aflatoxins synthesis, including 2-hydroxyisobutyrylation, succinylation, acetylation, and methylation (*Lv et al., 2022*; *Yang et al., 2019*; *Pfannenstiel et al., 2018*; *Ren et al., 2018*; *Wang et al., 2023b*; *Lv, 2017*; *Wang et al., 2022*), in which Snt2 (also called *sntB*, an epigenetic reader) is deeply involved. Despite advancements in the field, our understanding of the molecular mechanisms of aflatoxin production in *A. flavus* is still fragmentary.

The epigenetic reader encoded by *sntB* in *A. nidulans* was identified as a transcriptional regulator of the sterigmatocystin biosynthetic gene cluster and deletion of *sntB* gene in *A. flavus* results in loss of aflatoxin production (*Pfannenstiel et al., 2017*), increasing global levels of H3K9K14 acetylation and impairing several developmental processes (*Pfannenstiel et al., 2018*). The homolog gene in yeast, SNTB coordinates the transcriptional response to hydrogen peroxide stress (*Singh et al., 2012*; *Baker et al., 2013*). In *Penicillium expansum*, SntB regulated the development, patulin and citrinin production, and virulence on apples (*Tannous et al., 2020*). In *A. nidulans*, SntB combined with an H3K4 histone demethylase KdmB, a cohesin acetyltransferase (EcoA), and a histone deacetylase (RpdA) to form a chromatin binding complex (KERS) and bound to regulatory genes and coordinated fungal development with mycotoxin synthesis (*Karahoda et al., 2022*). In *A. flavus*, the KERS complex also consists of KdmB, RpdA, EcoA, and SntB plays a key role in the fungal development and secondary metabolites metabolism (*Karahoda et al., 2023*). SntB also regulated the virulence in *Fusarium oxysporum*, and respiration in *F. oxysporum* and *Neurospora crassa* (*Denisov et al., 2011a*; *Denisov et al., 2011b*).

However, the specific regulatory mechanism of SntB in *A. flavus* remains unclear. In this study, we identified the regulatory network of SntB by chromatin immunoprecipitation sequencing (ChIP-seq) and RNA sequencing (RNA-seq), which shed light on its impact on fungal biology.

## Results

### The phenotype of SntB in *A. flavus*

The role of SntB in *A. flavus* has been previously characterized by analyzing both Δ*sntB* and overexpression of *sntB* genetic mutants (*Pfannenstiel et al., 2018*). To further investigate the intrinsic mechanism of this regulator on the development and aflatoxin biosynthesis in *A. flavus*, the *sntB* deletion strain (Δ*sntB*) and the complementary strain (Com-*sntB*) were constructed by the method of homologous recombination and verified by diagnostic PCR (*Figure 1—figure supplement 1A*).

The expression levels of *sntB* in wild-type (WT), Δ*sntB*, and Com-*sntB* strains was further detected by quantitative RT-PCR (qRT-PCR) and the result showed that the expression of *sntB* was absent in the gene-deletion strain, and it fully recovered in the Com-*sntB* strain (*Figure 1—figure supplement 1B*), which reflected that the Δ*sntB* and Com-*sntB* strains had been successfully constructed, and could be used in the subsequent experiments of this study.

The phenotype analysis of this study revealed that the deletion of *sntB* gene significantly inhibited the growth of mycelium, hyphae morphology, the length of fungal cell (between two adjacent septa), the number of conidiation, sclerotium formation, and aflatoxin biosynthesis, while the above phenotypes of both development and mycotoxin biosynthesis were fully recovered in the Com-*sntB* strain (*Figure 1*). To reveal the signaling pathways of SntB in conidiation, sclerotium formation, and aflatoxin biosynthesis, qRT-PCR analysis was performed to assess the expression levels of sporulation-related transcriptional factor genes, *steA*, *wetA*, *fluG*, and *veA*, sclerotia formation-related transcriptional factor genes, *nsdC*, *nsdD*, and *sclR* (*Cary et al., 2012*; *Yang et al., 2018*), and the AFs synthesis gene cluster structural genes *aflC*, *aflR,* and *aflP*, and the main regulatory genes *aflR* and *aflS*. As shown in *Figure 1—figure supplement 2*, the relative expression levels of these genes were significantly lower in the Δ*sntB* strain compared to that of the WT strain, and recovered in the Com-*sntB* strain. These results indicated that SntB regulates the conidiation, sclerotium formation, and aflatoxin biosynthesis by the canonical signaling pathways mediated by these regulators.

## SntB plays important roles in virulence of *A. flavus* to both plant and animal hosts

In order to explore the effect of SntB on the fungal colonization ability, peanut seeds and maize kernels were infected with spore solution of each fungal strain. Compared to WT, the conidiation yield of Δ*sntB* on the infected host was significantly reduced (p<0.001) and no AFB1 could be detected on the hosts infected by Δ*sntB*, while in the Com-*sntB* strain, the capacity to produce conidia and AFB1 on both crop kernels was recovered (*Figure 2A–C* and *Figure 2—figure supplement 1A*). The role of *sntB* in fungal virulence to animals was also investigated. As shown in *Figure 2D and E*, the survival rate of silkworms injected by spores of Δ*sntB* strain was significantly higher than that from WT infected larvae. There was less fungal mycelium, conidia, and AFB1 production on the dead silkworms injected by Δ*sntB* compared to the silkworms from the WT injection groups, but when the gene was reintroduced (i.e. the Com-*sntB* group), similar to what found in the WT group, the survival rate of silkworms obviously dropped and more fungal mycelium, conidia, and AFB1 produced on the dead silkworms (*Figure 2F–H*). All the above results revealed that SntB plays an essential role in virulence of *A. flavus*.

The capacity of fungal infection is closely related to secreted hydrolases, such as amylase, lipase, protease, etc. In order to explore the effect of SntB on the activity of hydrolases, the activities of amylase in the above each fungal strain were further determined. The results showed that the colonies of Δ*sntB* produced almost no degradation transparent circle after adding iodine solution compared with that of WT and Com-*sntB*, which indicated that the activity of α-amylase in Δ*sntB* strain was significantly reduced (p<0.001) (*Figure 2—figure supplement 1B and C*). This suggests that SntB plays an important role in the fungal pathogenicity by changing the hydrolases activity of *A. flavus*.

## SntB chords global gene expression

To explore the downstream signaling pathways regulated by SntB, samples with three biological replicates of WT and Δ*sntB* strains were submitted for RNA-seq. In the assay, the bases score Q30 was more than 93.19% (*Supplementary file 1a*) and mapping ratio was from 95.17% to 95.80% (*Supplementary file 1b*). To further confirm the quality of RNA-seq, PCA and Pearson correlation analysis were performed. Correlation analysis revealed that the samples were clustered by groups (*Figure 3A*). A plot of PC1 (47.70%) and PC2 (23.20%) scores showed a clear separation between the groups (*Figure 3B*). A total of 1446 and 1034 genes were significantly up- and down-regulated, respectively, in the Δ*sntB* compared to the WT strain (*Figure 3C* and *Supplementary file 1c*). Gene ontology (GO) enrichment analysis identified 93 enriched GO terms (p<0.05) (*Figure 3D* and *Supplementary file 1d*). In the biological process category, the most enriched terms were 'oxidation-reduction process (GO: 0055114)'. In the molecular function category, 'catalytic activity (GO: 0003824),' 'oxidoreductase activity (GO: 0016491)', and 'cofactor binding (GO: 0048037)' were the most significantly enriched terms. Whereas terms associated with 'Set3 complex (GO: 0034967)', 'mitochondrial crista junction

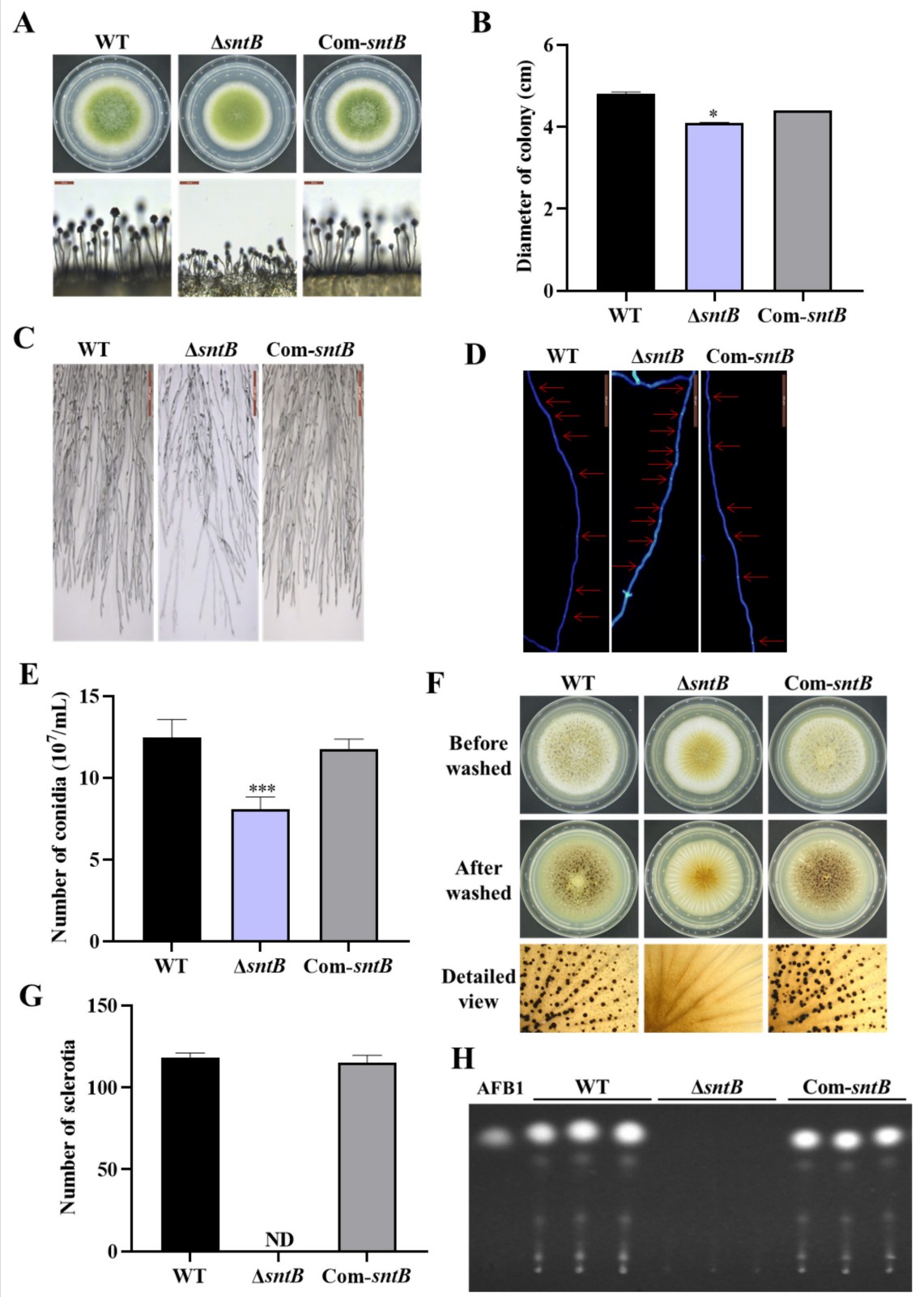

**Figure 1.** The functions of SntB in *A. flavus*. (**A**) The colonies of wild-type (WT), Δ*sntB*, and Com-*sntB* strains grown on potato dextrose agar (PDA) at 37°C in dark for 4 days. (**B**) The colony diameter statistics of the above fungal strains. (**C**) Microscopic examination revealed the difference in mycelia of each fungi strain at 37°C in dark, scale=200 μm. (**D**) Microscopic examination of the hyphal septum of each strain at 37°C in dark, scale=50 μm. (**E**) The spore production statistics. (**F**) All the above fungal strains were point-inoculated on CM medium and grown for 7 days at 37°C. (**G**) The number

*Figure 1 continued on next page*

*Figure 1 continued*

of sclerotia of the above fungal strains. ND=Not detectable. (**H**) AFB1 production of the above fungal strains was detected by TLC after the strains incubating at 29°C in PDB medium for 7 days.

The online version of this article includes the following source data and figure supplement(s) for figure 1:

**Figure supplement 1.** The construction of mutant strains.

**Figure supplement 1—source data 1.** Original files for PCR verification of gDNA in wild-type (WT), Δ*sntB,* and Com-*sntB* strains in *Figure 1—figure supplement 1*.

**Figure supplement 1—source data 2.** Original files for PCR verification of gDNA in wild-type (WT), Δ*sntB,* and Com-*sntB* strains in *Figure 1—figure supplement 1*, indicating the relevant bands and treatments.

**Figure supplement 2.** The expression of genes related to sporulation, sclerotia production, and aflatoxin synthesis.

---

(GO: 0044284)', and 'extracellular region (GO: 0005576)' were significantly enriched in the cellular component category. Additionally, all the DEGs were mapped according to the KEGG database, and 42 significantly enriched pathways were identified (p<0.05) (*Supplementary file 1e*). Among them, 'metabolic pathways (ko01100)', 'aflatoxin biosynthesis (ko00254)', and 'microbial metabolism in diverse environments (ko01120)' were the most significantly enriched (*Figure 3E* and *Supplementary file 1e*).

## Characterization of the binding regions of SntB by ChIP-seq

To characterize the chromatin regions targeted by SntB, ChIP-seq were carried out with both HA tag fused *sntB* strain (*sntB*-HA) and WT strain. The *sntB*-HA strain was constructed by homologous recombination through fused HA to the 3' end of *sntB* (*Figure 4A*). In the ChIP-seq assay, more than 94.66% of bases score Q30 and above in each sample (*Supplementary file 1f*), and reaching 52.50% to 94.48% of mapping ratio (*Supplementary file 1g*). The principal component analysis (PCA) (*Figure 4—figure supplement 1A*) and heatmap (*Figure 4—figure supplement 1B*) reflected that the quality of samples was competent for subsequent analysis. There were 1510 up-enriched differently accumulated peaks (DAPs) in *sntB*-HA fungal strain compared to the WT strain, which were distributed on the whole *A. flavus* genome (*Figure 4B* and *Supplementary file 1h*). Most of the up-enriched peaks were located in the promoter (82.85%) region (*Figure 4C*). To determine binding regions of SntB, we used the HOMER known and de novo motif discovery algorithm. Motifs were sorted based on p-values and the top 5 enriched known motifs were shown in *Figure 4D*. The results consisted of motifs derived from previously published ChIP-seq experiments on Cbf1, bHLHE40, NFY, Usf2, and USF1 (*Supplementary file 1i*). However, the most enriched de novo motif was NFYA (1e-97) (*Figure 4E*). The genes of the DAPs were further subjected to GO and KEGG analysis. The most strikingly enriched GO terms in the biological process category were 'cell communication (GO:0007154)', 'response to stimulus (GO:0050896)', and 'response to external stimulus (GO:0009605)'. Whereas terms associated with 'DNA-binding transcription factor activity (GO:0003700)', 'DNA-binding transcription factor activity, RNA polymerase II-specific (GO:0000981)', and 'sequence-specific DNA binding (GO:0043565)' were the most significantly enriched molecular function category (*Figure 4F* and *Supplementary file 1j*). And these genes were mostly enriched in 'Methane metabolism' and 'MAPK signaling pathway - yeast' pathways (p-value<0.05) (*Figure 4G* and *Supplementary file 1k*).

## Integration of the results of ChIP-seq and RNA-seq assays

After overlapping the results from both different sequence methods (ChIP-seq and RNA-seq), 238 DEGs were found (*Figure 5A*). According to the GO annotation, these DEGs were significantly enriched in eight GO terms, including 'cellular response to reactive oxygen species (GO:0034614)', 'reactive oxygen species metabolic process (GO:0072593)', and 'cellular response to oxygen-containing compound (GO:1901701)' (*Figure 5B*). It was further noted that the DEGs were significantly assigned to 'carbon metabolism (afv01200)', 'peroxisome (afv04146)', and 'glyoxylate and dicarboxylate metabolism (afv00630)' KEGG pathways (*Figure 5C* and *Supplementary file 1l and m*). These results revealed that SntB is essential for *A. flavus* to maintain the homeostasis of intracellular reactive oxygen species (ROS). Studies had shown that SNTB could response to oxidative stress in yeast (*Baker et al., 2013*) and *Magnaporthe oryzae* (*He et al., 2018*). As *Figure 5D* and *Figure 5E* showed, due to the deletion of the *sntB* gene, Δ*sntB* exhibited a severe menadione sodium bisulfite

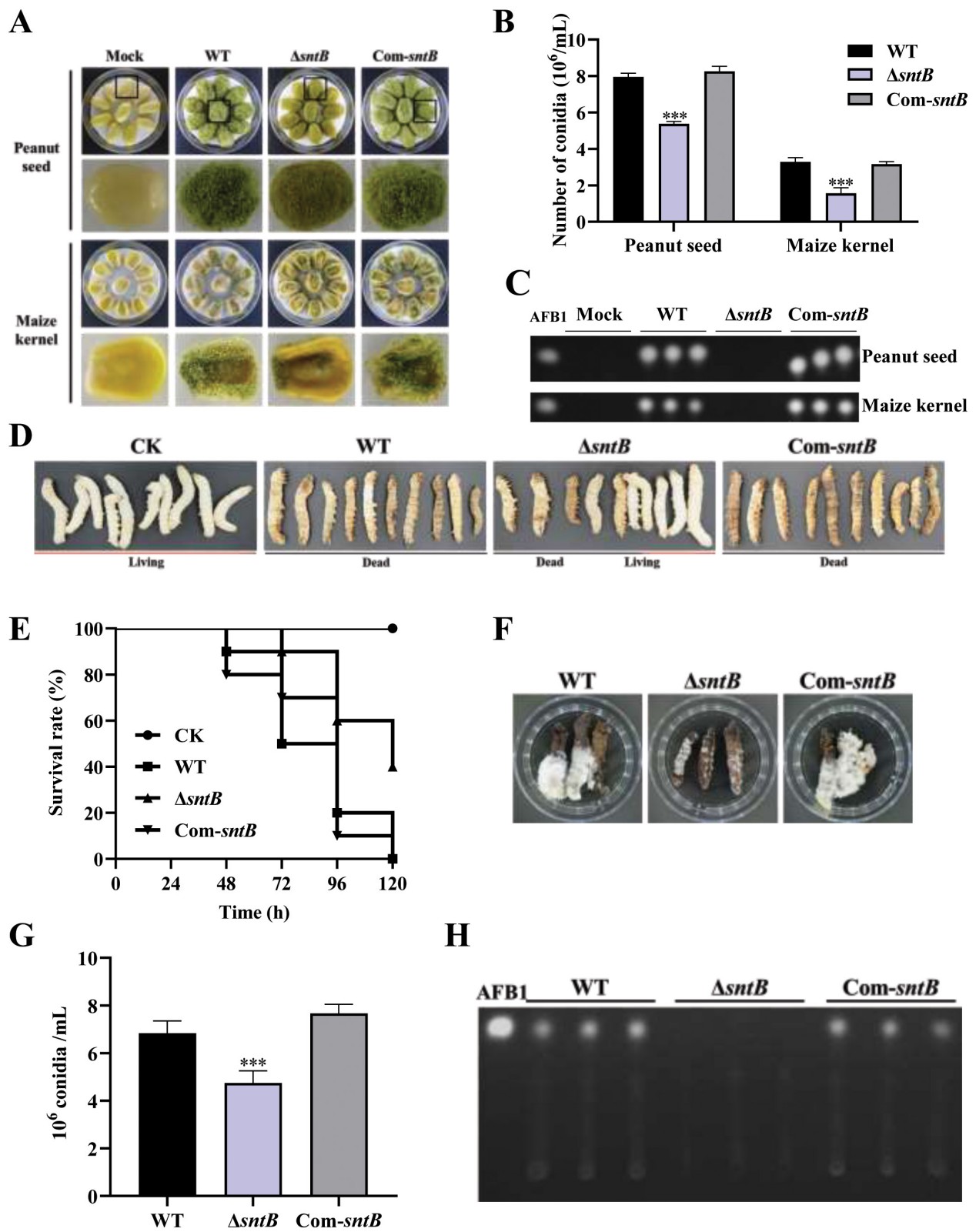

**Figure 2.** The role of SntB on the ability of *A. flavus* to colonize host. (**A**) Phenotype of peanut and maize kernels colonized by Δ*sntB*, Com-*sntB*, and wild-type (WT) strains at 29°C in dark for 7 days. (**B**) Statistical of the number of conidia on the surface of peanut and maize kernels. (**C**) TLC analysis to detect the yield of AFB1 in kernels infected by the above fungal strains after 7 days incubation. (**D**) Photographs of the silkworms infected by the above fungal strains. (**E**) The survival rate of silkworms in 5 days after injection of the above strains. (**F**) Photographs of the dead silkworms infected by *A. flavus*

*Figure 2 continued on next page*

*Figure 2 continued*

after 6 days incubation. (**G**) The spore production statistics of the above fungal strains on the dead silkworms shown in (**F**). (**H**) TLC analysis of AFB1 levels produced in infected dead silkworms in (**F**).

The online version of this article includes the following figure supplement(s) for figure 2:

**Figure supplement 1.** The changes of number of conidia, amylase, and lipase in wild-type (WT), Δ*sntB*, and Com-*sntB* strains.

(MSB) sensitivity phenotype compared to that of the WT strain, and the phenotype recovered in the complementary strain (Com-Δ*sntB*). The results showed that the inhibition rate of oxidant MSB to Δ*sntB* would be significantly enhanced with the increase of MSB concentration, which showed that SntB deeply participates in the regulation of oxidative stress pathway. As the most abundant peroxisomal enzyme, catalases (CAT) catalyze decomposition of hydrogen peroxide (*Okumoto et al., 2020*). To further study the mechanism of SntB-mediated oxidative response of *A. flavus*, the *catC* (encode a catalase) gene was selected based on the above integration results. According to the peak map, the binding region of SntB on *catC* gene was significantly enriched in *sntB*-HA strains compared to WT strain (*Figure 5F*). The motif in the binding region was shown in *Figure 5G* and the sequence was TCCGCCCG. The relative expression levels of *catC* in WT and Δ*sntB* strains under MSB treatment were measured. As shown in *Figure 5H*, the expression level of *catC* was significantly higher in Δ*sntB* strain than in WT strain, it suggested that to compensate the absence of *sntB*, *catC* is up-regulated to respond the higher intracellular oxidative level. However, under the stress of oxidant MSB, the deletion of *sntB* obviously suppressed the expression level of *catC* compared to that of WT strain, which reflected that the absence of *sntB* significantly impaired the capacity of *catC* to further respond to extra external oxidative stress. These results revealed that SntB is deeply involved in CatC-mediated oxidative response in *A. flavus*.

## CatC is important for *A. flavus* response to oxidative stress

The functions of *catC* gene in *A. flavus* were further explored by knockout of the *catC* (*Figure 6—figure supplement 1*). As shown in *Figure 6A–C*, the diameter of Δ*catC* strain was significantly smaller than that of WT, and the conidia number in the Δ*catC* strain decreased significantly compared to that of WT. The sclerotia production of Δ*catC* strain was also significantly less than that of WT strain (*Figure 6D and E*). In view of CatC is involved in the oxidative response pathways of *A. flavus* (*Figure 4*), both Δ*catC* and WT strains were treated by a serial concentration of MSB, and the results showed that the inhibition rates of MSB in Δ*catC* strain were significantly lower than that of WT (*Figure 6F and G*). Catalase is a major peroxisome protein and plays a critical role in removing peroxisome-generated ROS. The result of fluorescence intensity of oxidant-sensitive probe 6-carboxy-2',7'-dichlorodihydrofluorescein diacetate (DCFH-DA) showed that ROS accumulation in the Δ*catC* strain was higher than that in the WT strain (*Figure 6H*). This result echoed that the deletion of *sntB* increased intracellular oxidative level and the inhibition rate of MSB, and up-regulated the expression of *catC* (*Figure 5D–F*). The role of CatC in the biosynthesis of AFB1 was also assessed (*Figure 6I*). The results showed that a relatively large amount of AFB1 was produced by the Δ*catC* strain compared to the WT. But when under the stress of MSB, AFB1 yield of the WT strain was significantly more than that of Δ*catC* strain. All the above results revealed that the CatC plays an important role in SntB-mediated regulation pathway on fungal morphogenesis, oxidative stress responding, and AFB1 production.

## SntB regulates fungal virulence through peroxidase-mediated lipolysis

Biogenesis of peroxisomes was reported to promote lipid hydrolysis, increase the production of glycerol, and further change fungal pathogenicity (*Wang et al., 2023a*). Since it is deeply involved in oxidative response of *A. flavus*, we wondered if SntB also takes part in the regulation of the production of lipid and glycerol. As shown in *Supplementary file 1n*, one gene (G4B84_008359) in lipase activity GO term was significantly down-regulated in Δ*sntB* strain, which encoded a secretory lipase belonged to the virulence factors reported in *Pseudomonas aeruginosa* (*Papadopoulos et al., 2022*). The lipase activity was also assayed by examining the ability to cleave glycerol tributyrate substrate (*Sieber and Thummel, 2009*). The results showed that the colony diameter of Δ*sntB* strain on PDA medium with tributyrin were significantly smaller than that of the control, and the colony diameters of WT and Com-*sntB* strains on PDA medium were obviously bigger than those on 0.3% tributyrin PDA

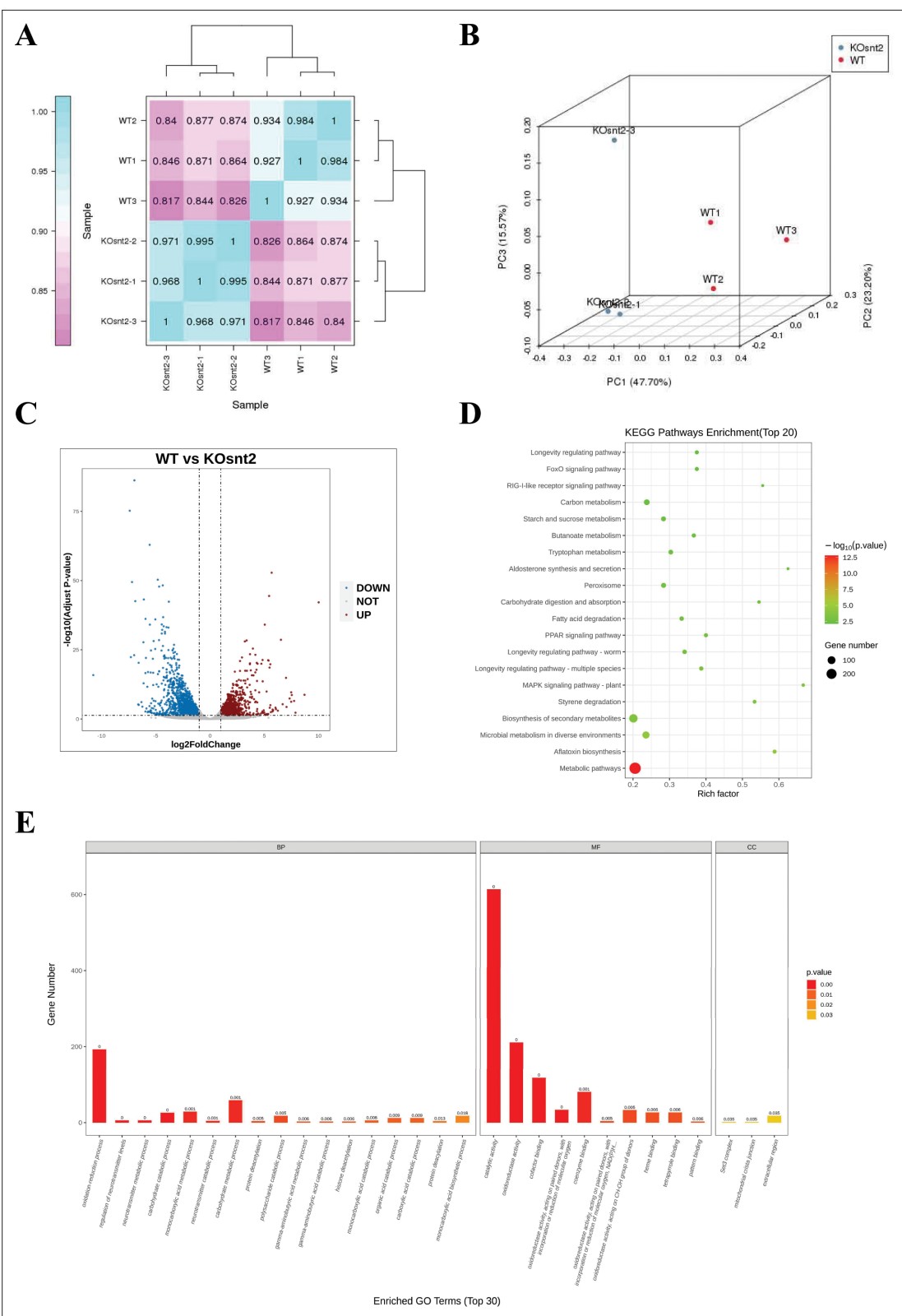

**Figure 3.** SntB chords global gene expression in *A. flavus*. (**A**) The Pearson correlation results shown by heatmap. (**B**) Principal component analysis (PCA) on six fungal samples, including three Δ*sntB* (KOsnt2) and three wild-type (WT) samples. (**C**) Volcano map reflecting the distribution of the differentially expressed genes. (**D**) Kyoto encyclopedia of genes and genomes (KEGG) analyses of the differentially expressed genes. (**E**) Gene ontology (GO) analyses of the differentially expressed genes.

*Figure 3 continued on next page*

*Figure 3 continued*

The online version of this article includes the following figure supplement(s) for figure 3:

**Figure supplement 1.** Heatmap of the differentially expressed genes (DEGs) related to oxidative response in transcriptome data drawn by TBtools.

medium. The relative inhibition rate of tributyrin to the colony growth of Δ*sntB* strains was significantly higher than that of WT and Com-*sntB* strain (*Figure 7A and B*). Our previous study revealed that H3 lysine 36 trimethylation (H3K36me3) modification on the chromatin region of the *sntB* is regulated by AshA and SetB (*Zhuang et al., 2022*). H3K36me3 usually promote gene transcription (*Zhao et al., 2019*; *Kooistra and Helin, 2012*). Our study revealed the potential machinery associated with SntB-mediated regulation on fungal morphogenesis, mycotoxin anabolism, and fungal virulence, which lurks the axle of from SntB to fungal virulence and mycotoxin biosynthesis through lipid catabolism (i.e. H3K36me3 modification-SntB-Peroxisomes-Lipid hydrolysis-fungal virulence and mycotoxin biosynthesis).

## Discussion

SntB is a conserved regulator in many species, including *Aspergillus nidulans* (*Pfannenstiel et al., 2017*), *Saccharomyces cerevisiae* (*Baker et al., 2013*), *Schizosaccharomyces pombe* (*Roguev et al., 2004*), *F. oxysporum* (*Denisov et al., 2011a*), *N. crassa* (*Denisov et al., 2011b*), and *A. flavus* (*Pfannenstiel et al., 2018*; *Pfannenstiel et al., 2017*). SntB can regulate the production of uncharacterized secondary metabolites, including aspergillicin A1 and aspergillicin F2 (*Greco et al., 2019*). MoSntB protein was required for regulation of infection-associated autophagy in *M. oryzae* (*He et al., 2018*). In *A. flavus*, the functions of *sntB* gene were previously analyzed by both Δ*sntB* and overexpression of *sntB* genetic mutants (*Pfannenstiel et al., 2018*). SntB deletion impaired several developmental processes, such as sclerotia formation and heterokaryon compatibility, secondary metabolite synthesis, and ability to colonize host seeds, which were consistent with our results (*Figures 1 and 2*). Unlike this, a complementation strain was constructed in this study which further clarified and confirmed the function of *sntB* gene. In this study, the potential mechanism under these effects was further analyzed by the detection of the related transcriptional factor genes of sporulation (*steA*, *WetA*, *fluG*, and *veA*), sclerotia formation (*nsdC*, *nsdD*, and *sclR*), and the AFs synthesis-related genes *aflC*, *aflD*, *aflO*, *aflP*, and *aflR* (*Figure 1—figure supplement 2*). In the RNA-seq data, we also found some DEGs related to AFs synthesis (*aflB*, *aflE*, *aflH*, *aflK*, *aflN*, *aflO*, *aflP*, *aflQ*, *aflR*, *aflS*, *aflV*, and *aflW*) (*Figure 3—figure supplement 1*). And all these genes were down-regulated, which was consistent with that the AFs production in Δ*sntB* was significantly decreased compared to WT and Com-*sntB* (*Figure 1F*). These results inferred that SntB regulated the morphogenesis and the production of *A. flavus* through the above canonical signal pathways.

For the process of *A. flavus* invading hosts, in view of it is a notorious pathogen for plant and animal, we established both crop and insect models, especially silkworm represented animal mode profoundly revealed the critical role of SntB in fungal virulence (*Figure 2*). The results of crop kernel models showed that the number of spores of Δ*sntB* on kernels of both peanut and maize was dramatically lower than that of strains WT and Com-*sntB* (*Figure 2A and B*) and almost no AFB1 was detected on maize and peanut kernels inoculated with Δ*sntB*, while plenty of AFB1 were detected from the kernels infected by WT fungal strain, and AFB1 biosynthesis capacity of Com-*sntB* strains recovered compared to the Δ*sntB* and WT fungal strains (*Figure 2C*). These results were corroborated by previous study (*Pfannenstiel et al., 2018*). It was also found in this study that the survival rate of silkworms injected by spores of Δ*sntB* strain was significantly higher than the silkworms injected with spores from WT and Com-*sntB* fungal strains (*Figure 2D and E*). What's more, we also assayed the effect of SntB on the activity amylase, which was closely related to the capacity of fungal infection (*Mellon et al., 2007*). As shown in *Figure 2—figure supplement 1C*, after adding iodine solution, the Δ*sntB* strain almost did not produce a degradation transparent ring compared to WT and complementary strains, indicating the amylase activity of the *sntB* gene knockout strain was significantly decreased (p<0.001). Our results comprehensively reveal the important function of SntB in the growth, development, secondary metabolite synthesis, and virulence of *A. flavus*.

SntB was reported as an important epigenetic reader. In *A. flavus*, SntB was reported to regulate global histone modifications (acetylation and methylation) and interact with EcoA and RpdA to form

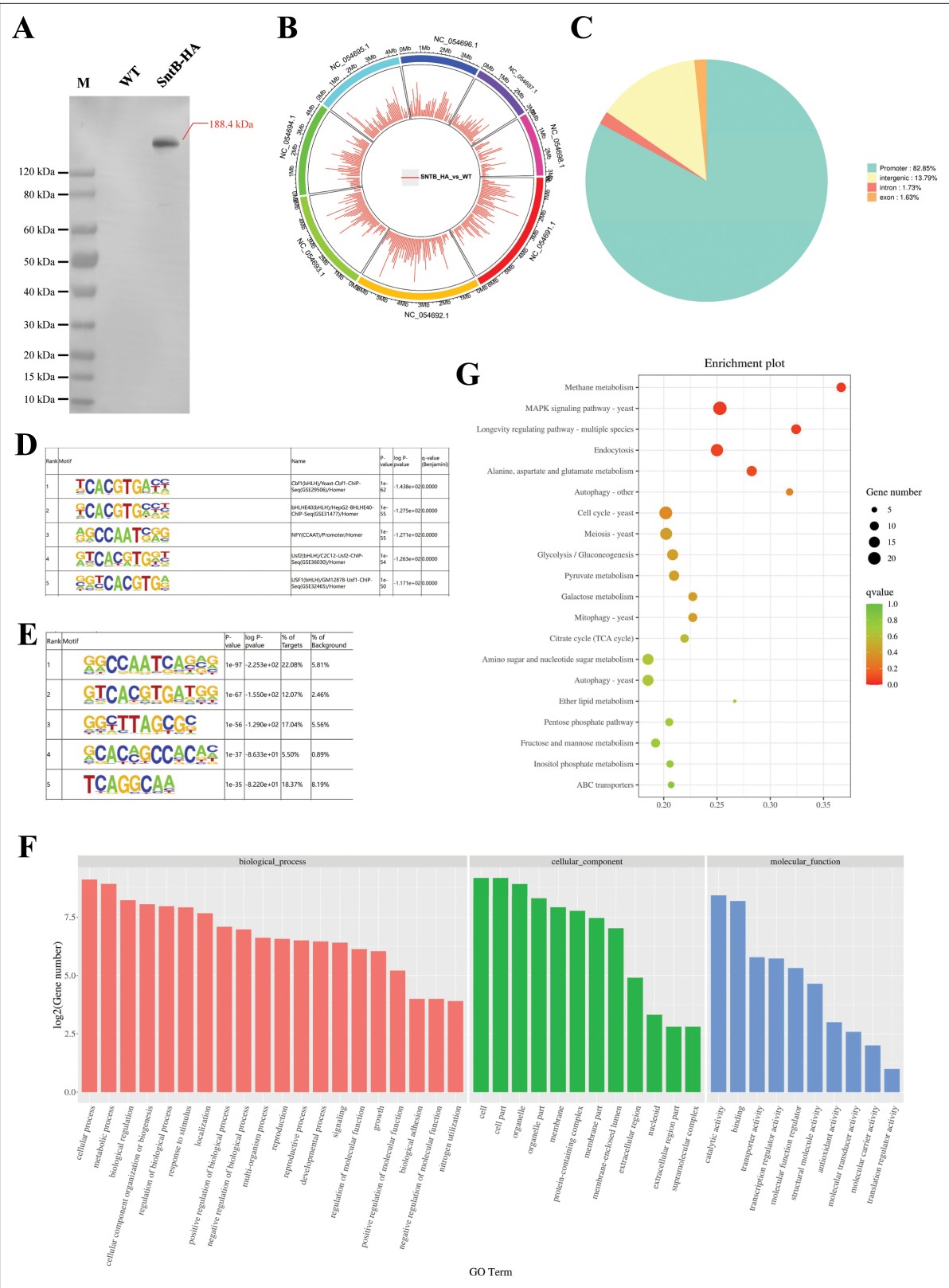

**Figure 4.** Characterization of the binding regions of SntB. (**A**) Verification of the construction of *sntB*-HA strain using western blot. M means the protein marker of PAGE-MASTER Protein Standard Plus (GenScript USA, MM1397). (**B**) The distribution of differently accumulated peaks on the genome. (**C**) Vennpie map of the differently accumulated peaks distribution on gene functional elements. (**D**) Enrichment of known motifs showing the top-ranked

*Figure 4 continued on next page*

*Figure 4 continued*

motif logos. (**E**) Enrichment of de novo motifs showing the top-ranked motif logos. (**F**) Gene ontology (GO) analyses of the differently accumulated peak-related genes. (**G**) Kyoto encyclopedia of genes and genomes (KEGG) analyses of the differently accumulated peak-related genes.

The online version of this article includes the following source data and figure supplement(s) for figure 4:

**Source data 1.** Original files for western blot analysis displayed in *Figure 4A*.

**Source data 2.** Original files for western blot analysis displayed in *Figure 4A*, indicating the relevant bands and treatments.

**Figure supplement 1.** Sequence information of chromatin immunoprecipitation sequencing (ChIP-seq).

a conserved chromatin regulatory complex (*Pfannenstiel et al., 2018*; *Karahoda et al., 2022*). Loss of *sntB* in *M. orzyae* also led to an increase in H3 acetylation (*He et al., 2018*). In our RNA-seq data, we also found a set domain containing histone-lysine *N*-methyltransferase (Ash1, G4B84_009862) was down-regulated in Δ*sntB* strain compared to WT (*Supplementary file 1c*), which was reported to regulate mycotoxin metabolism and virulence via H3K36 methylation in *A. flavus* (*Zhuang et al., 2022*). Besides, SntB is reported to be a transcriptional regulator in *A. nidulans* (*Pfannenstiel et al., 2017*) and *F. oxysporum* (*Denisov et al., 2011a*). So, we used RNA-seq and ChIP-seq to study the transcriptional response of *sntB* in *A. flavus*. By integration analysis of the results of ChIP-seq and RNA-seq assays, we found that the enriched GO terms and KEGG pathways of the DEGs were related to oxidative response (*Figure 5A–C*). These results reflected that SntB plays an important role in fungal response to oxidative stress, which is consistent with the previous reports that SntB could respond to oxidative stress in yeast (*Baker et al., 2013*), *F. oxysporum* (*Denisov et al., 2011a*), and *M. oryzae* (*He et al., 2018*).

As a harmful by-product of oxidative metabolism, ROS is unavoidable and essential for fungus development (*Aguirre et al., 2005*; *Gessler et al., 2007*). ROS has also been shown to be required for aflatoxin production (*Jayashree and Subramanyam, 2000*; *Roze et al., 2011*; *Yang et al., 2015*). Several oxidative stress-responsive transcription factors have been identified as regulating aflatoxin production, including AtfB, AP-1, and VeA (*Roze et al., 2011*; *Reverberi et al., 2008*; *Sakamoto et al., 2008*; *Baidya et al., 2014*). Previous studies have shown that SntB protein coordinates the transcriptional response to hydrogen peroxide-mediated oxidative stress in the yeast (*Singh et al., 2012*; *Baker et al., 2013*) and is involved in fungal respiration and ROS in *F. oxysporum* and *N. crassa* (*Denisov et al., 2011a*; *Denisov et al., 2011b*). Several GO terms ('cellular response to reactive oxygen species', 'reactive oxygen species metabolic process', and 'cellular response to oxygen-containing compound') and KEGG pathways (peroxisome) were enriched by the DEGs screened out from integration of ChIP-seq and RNA-seq data in this study (*Figure 5*). And the intracellular ROS level in the Δ*sntB* and Δ*catC* strains was significantly higher than that in WT strain (*Figure 6H*), which was similar to previous report on the *cat1* gene in *A. flavus* (*Zhu et al., 2020*). This is the first time to show that the SntB in *A. flavus* is important in oxidative stress response, through which SntB participates in the regulation of aflatoxin biosynthesis and fungal development.

Fungal defense against ROS is mediated by superoxide dismutases (SOD), CAT, and glutathione peroxidases. The effect of MSB on cellular growth and antioxidant enzyme induction in *A. flavus* was previously explored (*Wang et al., 2020*; *Vig et al., 2023*; *Zaccaria et al., 2015*). Once in the cell, menadione may release superoxide anion (*Criddle et al., 2006*), which was scavenged by SOD and transformed into hydrogen peroxide, or react with nitric oxide to form peroxynitrite (*Ferreira et al., 2013*). This study found that after knocking out *sntB* gene, the strain growth was significantly inhibited by MSB (*Figure 5D and E*). Some genes encoded SOD and CAT were reported to be associated with AF/ST synthesis (*Caceres et al., 2020*), including *mnSOD*, *sod1*, *sod2*, *catA*, *catB*, and *hyr1*. In our RNA-seq data, seven related genes were screened out (*Supplementary file 1o*), among which bZIP transcription factor *Atf21* (G4B84_008675), *catlase C* (G4B84_009242), *catlase A* (G4B84_010740), superoxide dismutases *sod2* (G4B84_003204), peroxisomal membrane protein *PmpP24* (G4B84_001452) were up-regulated, while *catlase B* (G4B84_008381) and superoxide dismutase *sod1* (G4B84_009129) were down-regulated in Δ*sntB* strain, respectively. The binding region of SntB on *catA*, *catB*, *sod1*, *sod2*, and *catC* genes promoter was significantly enriched in *sntB*-HA strain compared to WT strain (*Figure 5F* and *Figure 6—figure supplement 2A*). Also, the motif in the binding regions was shown in *Figure 5G* and *Figure 6—figure supplement 2B*. These findings suggest that SntB has the ability

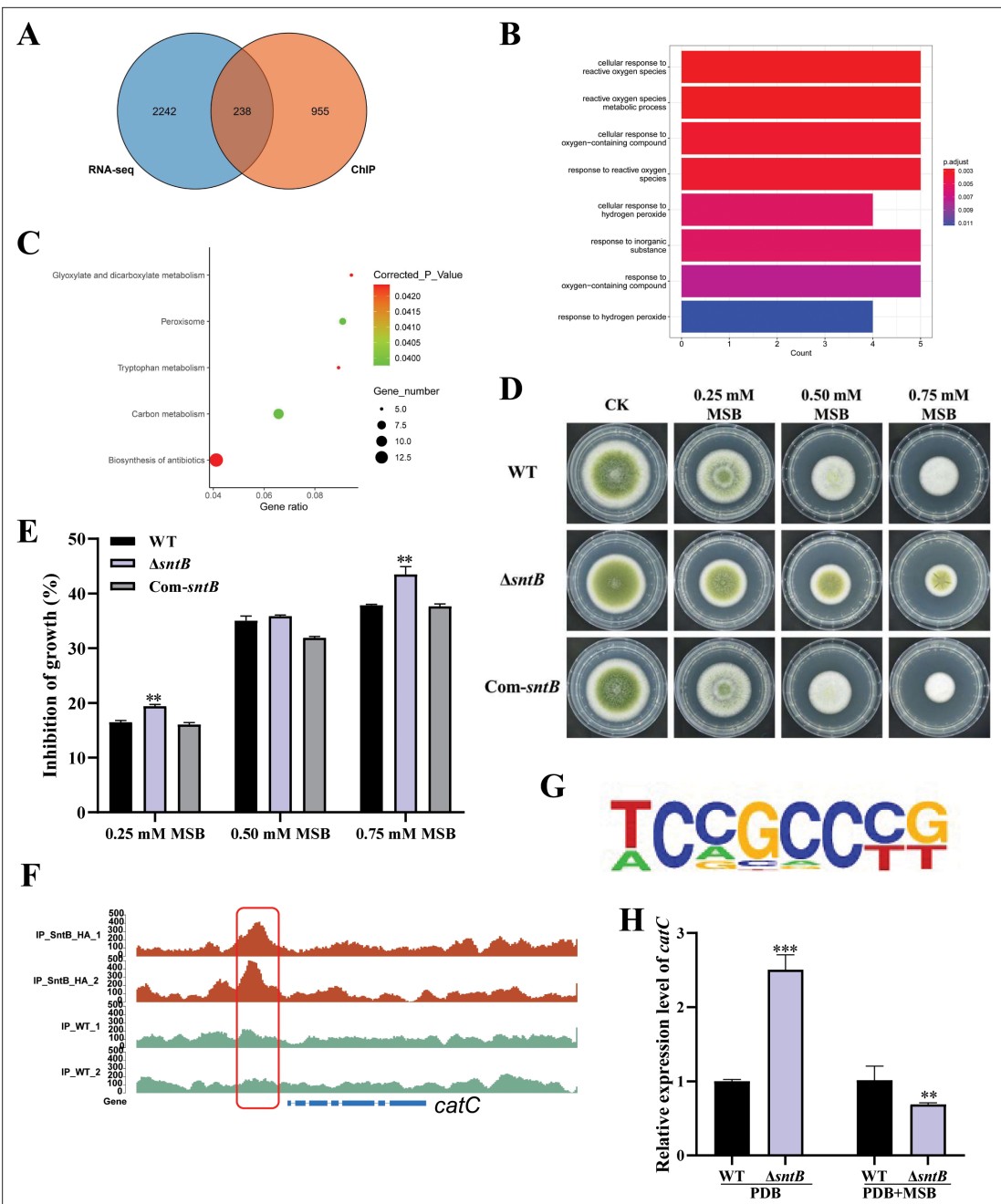

**Figure 5.** Integration of the results of chromatin immunoprecipitation sequencing (ChIP-seq) and RNA sequencing (RNA-seq) assays. (**A**) Venn diagrams of ChIP-seq and RNA-seq. (**B**) Gene ontology (GO) analyses of the common genes. (**C**) Kyoto encyclopedia of genes and genomes (KEGG) analyses of the common genes. (**D**) The phenotype of wild-type (WT), ΔsntB, and Com-sntB strains cultured in PDA containing a series concentration of menadione sodium bisulfite (MSB) for 3 days. (**E**) Statistical analysis of the growth inhibition rate of MSB to all the above fungal strains according to Panel D. (**F**) Comparison of the enrich levels of the SntB binding region of catC gene between WT and sntB-HA strains. (**G**) The motif logo in the SntB binding region of catC gene. (**H**) The relative expression level of catC in WT and ΔsntB strains with or without MSB treatment.

to interact with the promoter region of genes associated with redox processes, thereby modulating the expression of these genes.

Some studies reported the correlation among ROS formation, aflatoxin production, and antioxidant enzyme activation. Aflatoxin B1 biosynthesis and the activity of total SOD were effectively inhibited by cinnamaldehyde, whereas the activities of catalase and glutathione peroxidase were opposite (*Sun et al., 2016*). The expression of *catA, cat2,* and *sod1,* and CAT enzymatic activity were opposite

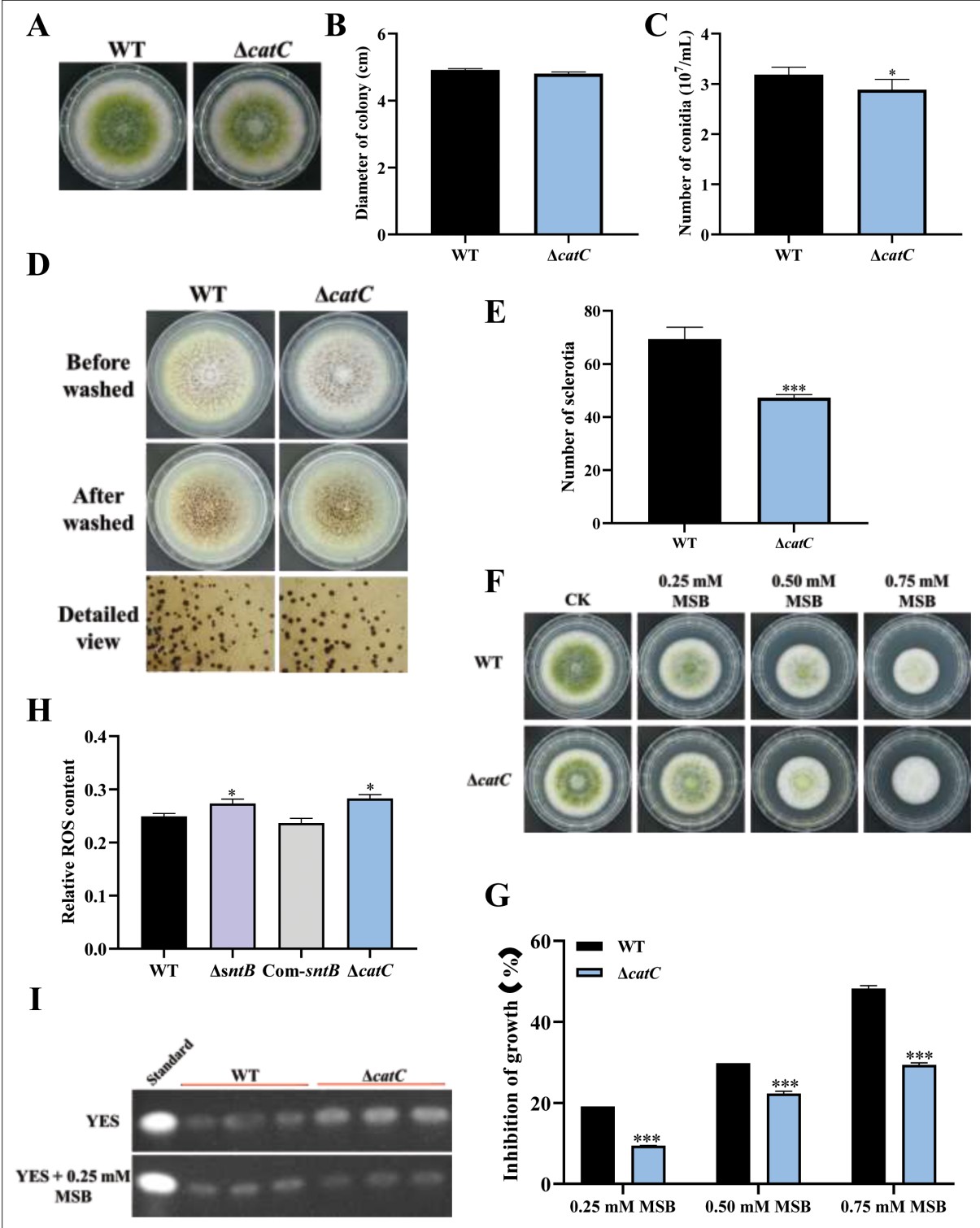

**Figure 6.** The functions of *catC* in *A. flavus*. (**A**) The colonies of wild-type (WT) and Δ*catC* strains grown on potato dextrose agar (PDA) at 37°C in dark for 4 days. (**B**) The colony diameter statistics of the above fungal strains. (**C**) The spore production statistics of the above fungal strains. (**D**) All above fungal strains were point-inoculated on complete medium (CM) and grown for 7 days at 37°C. (**E**) The number of sclerotia of the above fungal strains. (**F**) The phenotype of above strains cultured on PDA medium containing a series concentration of menadione sodium bisulfite (MSB) for 3 days. (**G**) Statistical analysis of the growth inhibition rate of MSB to all the above fungal strains according to (**F**). (**H**) Relative reactive oxygen species (ROS) levels in the WT, Δ*sntB*, Com-sntB, and Δ*catC* strains. (**I**) AFB1 production of the above fungal strains was detected by TLC after the strains incubating at 29°C in potato dextrose (PDB) medium for 7 days.

*Figure 6 continued on next page*

*Figure 6 continued*

The online version of this article includes the following source data and figure supplement(s) for figure 6:

**Source data 1.** Original files for TLC detection of AFB1 production displayed in *Figure 6I*.

**Source data 2.** Original files for TLC detection of AFB1 production displayed in *Figure 6I*, indicating the relevant bands and treatments.

**Figure supplement 1.** PCR verification of gDNA in wild-type (WT) and Δ*catC*.

**Figure supplement 1—source data 1.** Original files for PCR verification of gDNA in wild-type (WT), Δ*catC* strains in *Figure 6—figure supplement 1*.

**Figure supplement 1—source data 2.** Original files for PCR verification of gDNA in wild-type (WT), Δ*catC* strains in *Figure 6—figure supplement 1*, indicating the relevant bands and treatments.

**Figure supplement 2.** The binding region and motif of SntB on the *catA*, *catB*, *sod1*, and *sod2* genes.

correlated to AFB1 biosynthesis under AFs inhibitor piperine treatment (*Caceres et al., 2017*). Deletion of the gene *sod* (GenBank accession no: CA747446) reduced AFs production (*He et al., 2007*), which was most similar to *sod2* (G4B84_003204). The mitochondria-specific *sod* and the genes *aflA*, *aflM*, and *aflP* belonging to the AFs gene cluster were reported to be co-regulated (*Hong et al., 2013*). Ethanol can inhibit fungal growth and AFB$_1$ production in *A. flavus* and enhanced levels of antioxidant enzymatic genes, including *Cat*, *Cat1*, *Cat2*, *CatA*, and Cu, Zn SOD gene *sod1*. All these reports indicated that the expression of antioxidant enzymatic genes was opposite correlated to AFB1 biosynthesis.

In our study, seven genes related to oxidative response were obviously differentially expressed in transcriptome data (*Supplementary file 1o*). Among these DEGs, five out of seven genes were up-regulated in Δ*sntB* strain. Based on the AFs production in Δ*sntB* was significantly decreased compared to WT and Com-*sntB* (*Figure 1F*), the most up-regulated gene in Δ*sntB* strain, *catC* (G4B84_000242), was selected for further analysis. We found that the deletion of *sntB* significantly up-regulated the *catC* gene, however, the expression of the *catC* gene was suppressed under MSB treatment (*Figure 5H*). Results also showed that the inhibition rates of MSB to Δ*catC* strain was significantly lower than that of WT group and AFB1 yield of the Δ*catC* strain was significantly decreased than that of WT strain under the stress of MSB (*Figure 6F–I*). These results indicated that SntB is profoundly involved in the CatC-mediated oxidative stress sensitivity response.

Peroxisomes are intimately associated with the metabolism of lipid droplets (*Joshi and Cohen, 2019*) and the histone lysine methyltransferase ASH1 promotes peroxisome biogenesis, inhibits lipolysis, and further affects pathogenesis of *Metarhizium robertsii* (*Wang et al., 2023a*). Set2 histone methyltransferase family in *A. flavus*, AshA and SetB, were found to regulate mycotoxin metabolism and virulence via H3K36me3, including the chromatin region of the *sntB* (*Zhuang et al., 2022*). By ChIP-seq and RNA-seq, SntB was found to be essential for *A. flavus* to maintain the homeostasis of intracellular ROS (*Figure 5A*) and several antioxidant enzymes were up-regulated in Δ*sntB* strain (*Supplementary file 1o*). In addition, we also found only one down-regulated DEG (G4B84_008359) in lipase activity GO term in our RNA-seq data (*Supplementary file 1n*), which encodes a secretory lipase and belongs to the virulence factors reported in *P. aeruginosa* (*Papadopoulos et al., 2022*). These results suggested that SntB plays a pivotal role in regulating peroxisome biogenesis to promote lipolysis involving in fungal pathogenesis.

Overall, we explored and clarified the bio-function of the SntB and found that SntB responses to oxidative stress through related oxidoreductase represented by CatC in *A. flavus* (*Figure 7*). Our study revealed the potential machinery associated with SntB-mediated regulation on fungal morphogenesis, mycotoxin anabolism, and fungal virulence, which lurks the axle of from SntB to fungal virulence and mycotoxin biosynthesis (i.e. SntB-Peroxisomes-Lipid hydrolysis-fungal virulence and mycotoxin biosynthesis). The work of this study provided a novel perspective for developing new prevention and control strategies against pathogenic fungi.

## Materials and methods

### *A. flavus* strains, media, and culture conditions

*A. flavus* Δ*ku70* Δ*pyrG* was used as a host strain, for genetic manipulations. All strains used in this study are listed in *Table 1*. Potato dextrose agar (PDA, 39 g/L, BD, Difco, Franklin, NJ, USA), complete medium (CM, 6 g/L tryptone, 6 g/L yeast extract, 10 g/L glucose), and potato dextrose (PDB, 24 g/L,

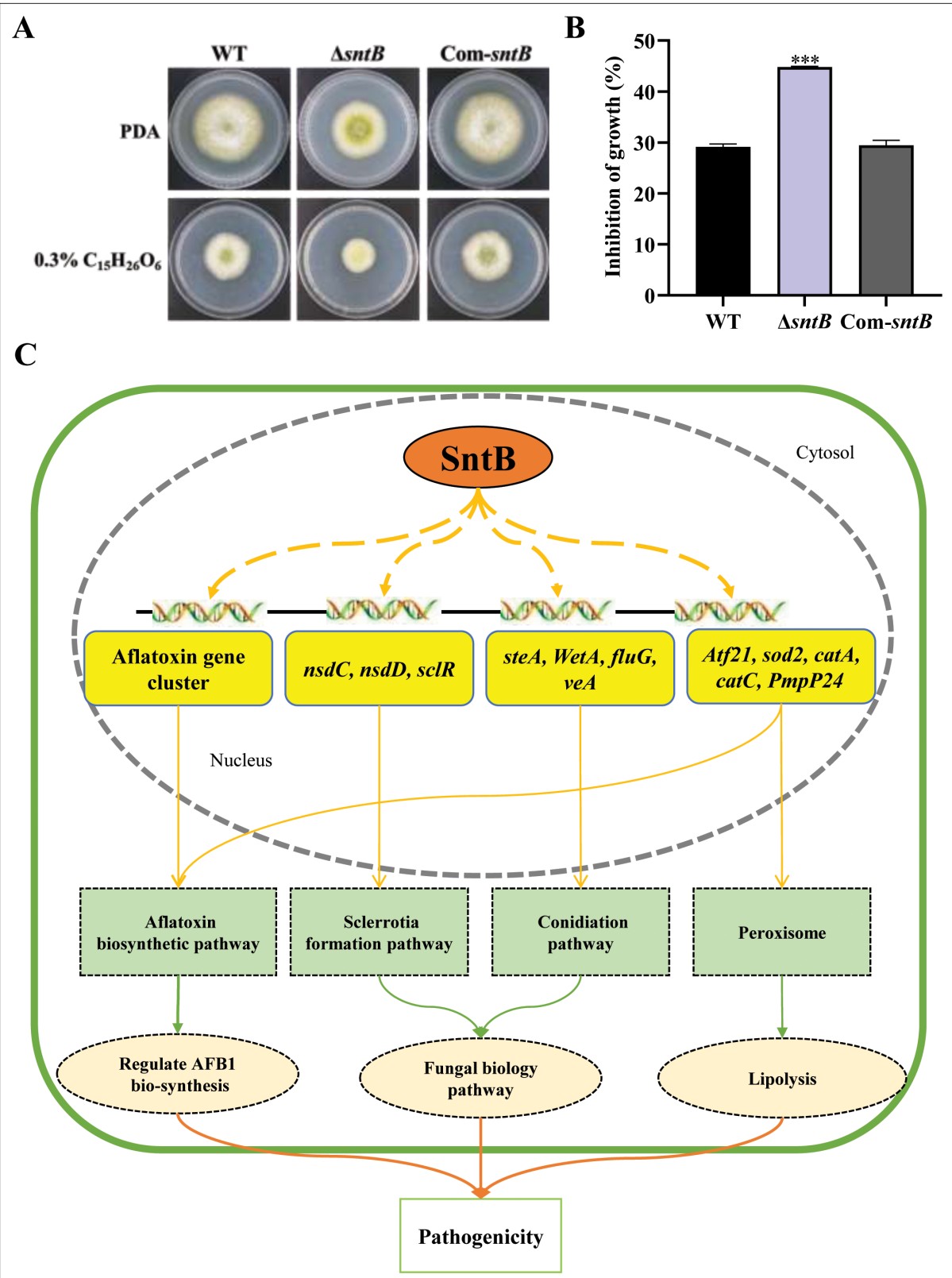

**Figure 7.** SntB regulate peroxisome biogenesis, fatty acid utilization, and fungal pathogenicity in *A. flavus*. (**A**) The phenotype of each strain on PDA medium containing 0.3% tributyrin. (**B**) Statistics of inhibition rates. The asterisk *** above the bars represents significantly different (p<0.001). (**C**) Mechanistic diagram of the bio-functions of SntB in *A. flavus*.

BD, Difco, Franklin, NJ, USA) were used for mycelial growth and sporulation determination, sclerotia production, and mycotoxin production analysis, respectively. All experiments were technically repeated three times and biologically repeated three times.

## The construction of mutant strains

All mutant strains, including *sntB* and *catC* gene knock-out strain (Δ*sntB* and Δ*catC*), the complementation strain for the Δ*sntB* strain (Com-*sntB*), and HA tag fused to *sntB* strain (*sntB*-HA), were constructed following the protocol of homologous recombination (*Zhuang et al., 2022*) and the detail protocol was as described in our previous study (*Pan et al., 2023*). The related primers were listed in *Table 2*. The constructed strains were confirmed by diagnostic PCRs (*Hu et al., 2018*). The construction of *sntB*-HA was further determined by western blot with anti-HA antibody (MCE, YA3393) as described previously (*Pan et al., 2023*).

## Phenotypic analysis and aflatoxin analysis

The spores ($10^7$ conidia/mL) of WT, Δ*sntB*, and Com-*sntB* strains were used. The details of the experiment were according to our previous study (*Pan et al., 2023*). Hyphal septum was stained according to the described method (*Kaminskyj, 2000*). Each fungal strain was evaluated on four plates, and each experiment was repeated three times.

## Fungal colonization on crop kernels

According to our previous experimental protocol (*Pan et al., 2023*), the colonizing ability of WT, Δ*sntB*, and Com-*sntB* fungal strains on peanut and corn kernels was analyzed. The crop kernels were disinfected with 0.05% sodium hypochlorite and soaked for 30 min in a solution containing $10^5$ conidia/mL fungal spores. Afterward, the seeds were placed in a Petri dish and cultured at 29°C for 6 days. Finally, the number of conidia was calculated and AFB1 product was analyzed by TLC.

## Animal invasion assay

The animal invasion assay using silkworms (*Bombyx mori*) was conducted according to our previous study (*Zhuang et al., 2022*; *Zhang et al., 2021*). Silkworms were randomly separated into four groups (10 larvae/group) when silkworm larva reach about 1 g in weight. Each silkworm was injected with 5 μL saline, or 5 μL conidial suspension ($10^6$ spores/mL) from WT, Δ*sntB*, and Com-*sntB* strains. The survival rate of silkworms was calculated. Dead silkworms were transferred into fresh 9 cm Petri dishes and cultivated for 5 days in the dark. The conidia number and AFB1 production from each group were measured.

## RNA-seq analysis

To reveal the potential complex regulatory network of the *sntB*, RNA-seq analysis was carried out on the WT and Δ*sntB* strains by Applied Protein Technology, Shanghai (https://www.aptbiotech.com) (*Wen et al., 2022*). Data processing was according to a previous study (*Hao et al., 2023*). Differentially expressed genes (DEGs) were assigned as genes with |log2FoldChange|>1 and adjusted p-adj<0.05. Gene Ontology (GO) and Kyoto Encyclopedia of Genes and Genomes (KEGG) pathways were used to analyze the functions of DEGs.

**Table 1.** *A. flavus* strains used in this study.

| Strain name | Related genotype | Source |
|---|---|---|
| *A. flavus* CA14 | Δ*ku70*, Δ*pyrG* | Kindly presented from Prof. Chang (*Chang et al., 2010*) |
| Wild-type (WT) | Δ*ku70*, Δ*pyrG::AfpyrG* | This study |
| Δ*sntB* | Δ*ku70*, Δ*sntB::AfpyrG* | This study |
| Com-*sntB* | Δ*ku70*, Δ*pyrG*; Δ*sntB::AfpyrG:: sntB* | This study |
| *sntB*-HA | Δ*ku70*, *sntB*-HA::*AfpyrG* | This study |
| Δ*catC* | Δ*ku70*, Δ*catC::AfpyrG* | This study |

**Table 2.** Primers used for strain construction in this study.

| Primer name | Sequence (5' → 3') | Fragment amplified |
|---|---|---|
| *sntB*-p1 | CTTCTCGAATTCCCCTTCATGACACTCTCC | For the construction of knock-out strain |
| *sntB*-p2 | GCTAAATCAGGATGGGTTGGAGGGTGAC | |
| *sntB*-p3 | GCATCTCCTTTGTGTTGTTTGGACCGTGT | |
| *sntB*-p4 | CAACCAACCACTGACGTCGACCAC | |
| *sntB*-p5 | GTCACCCTCCAACCCATCCTGATTTAG CGCCTCAAACAATGCTCTTCACCC | |
| *sntB*-p6 | ACACGGTCCAAACAACACAAAGGAGAT GCGTCTGAGAGGAGGCACTGATGC | |
| *sntB*-p7 | TAGATCACCCAGCGGGCCACAA | |
| *sntB*-p8 | GACTCAAATGGAAATCCCGTCGTGCC | |
| *sntB*-F | GCAGCAACACCACGTGAGGCCCAATTC | A fragment from *sntB* ORF |
| *sntB*-R | CCAGGTCACAGGGCATAGAACACACTCGTA | |
| P1020-F | ATCGGCAATACCGTCCAGAAGC | Verify the fragment of *AfpyrG* |
| P801-R | CAGGAGTTCTCGGGTTGTCG | |
| *sntB*-C-p2 | GGGTGAAGAGCATTGTTTGAGG CCCTCCAACCTTACTCCGTACACAATTCTAG | For the construction of complementary strain of *sntB* |
| s*ntB*-C-p3 | GCATCAGTGCCTCCTCTCAGACG GCACGACGGGATTTCCATTTGAGTC | |
| *sntB*-C-p4 | CCAATTTCCTGATGATTGTGATGTGTGTCC | |
| SC-P-F | GCCTCAAACAATGCTCTTCACCC | |
| SC-P-R | GTCTGAGAGGAGGCACTGATGC | |
| sntB-HA-P1 | GAGATTTATCGACGACTATATGGC | For the construction of HA tag fused strain of *sntB* |
| sntB-HA-P2 | TGAACCTCCGCCACCACTACCTCCGCCACCAG AGAGTAAATTCTTGAGAGATGGG | |
| HA-linker-pyrG-F | TGGCGGAGGTAGTTACCCATACGACGTCCCAGACTA CGCTTACCCATACGACGTCCCAGACTACGCTTACCCA TACGACGTCCCAGACTACGCTTGAGCCT CAAACAATGCTCTTCACCC | |
| *catC*-P1 | CTTGAGACGCAGGACGAA | For the construction of knock-out strain of *catC* |
| *catC*-P2 | GGGTGAAGAGCATTGTTTGAGGCT GATGTGGGTTGTATGAATG | |
| *catC*-P3 | GCATCAGTGCCTCCTCTCAGACTG GATGCGGGTGAATACTG | |
| *catC*-P4 | ACAAGCTGTCATGCGTGG | |
| *catC*-P5 | TGGGAGTCTCGAACACAC | |
| *catC*-P6 | GAAAACCCCGCAACAGAC | |
| *catC*-F | AGCCTATTTCGGACCCCT | |
| *catC*-R | CAGTCTCCTTTCGGCATC | |

## ChIP-seq and data analysis

ChIP-seq analysis was carried out on the WT and *sntB*-HA strains. The conidia ($10^4$/mL) of each strain were inoculated in 100 mL PDB shaking at 180 rpm under 29°C for 72 hr, and subjected to ChIP-seq analysis by Wuhan IGENEBOOK Biotechnology Co., Ltd (http://www.igenebook.com). ChIP experiment was carried out according to a previous study (*Zhuang et al., 2022*). Raw sequencing with low-quality reads were discarded, and reads contaminated with adaptor sequences trimmed were filtered

**Table 3.** Primers used for RT-qPCR in this study.

| Primer name | Sequence (5′ → 3′) | Fragment amplified |
|---|---|---|
| sntB-qF | ACTCATCAGAGCCCCTATGGGCCAGTC | |
| sntB-qR | GCACTAAGAACGCGATCGACAGAATAGACAC | |
| catC-qF | GAAAGAGTTGTCCATGCCA | |
| catC-qR | CAGAAAACGGGTGTGTGAT | |
| brlA-F | GCCTCCAGCGTCAACCTTC | |
| brlA-R | TCTCTTCAAATGCTCTTGCCTC | |
| abaA-F | TCTTCGGTTGATGGATGATTTC | |
| abaA-R | CCGTTGGGAGGCTGGGT | |
| nsdC-F | GCCAGACTTGCCAATCAC | |
| nsdC-R | CATCCACCTTGCCCTTTA | |
| nsdD-F | GGACTTGCGGGTCGTGCTA | Sclerotium-related genes |
| nsdD-R | AGAACGCTGGGTCTGGTGC | |
| sclR-F | CAATGAGCCTATGGGAGTGG | |
| sclR-R | ATCTTCGCCCGAGTGGTT | |
| aflC-F | GTGGTGGTTGCCAATGCG | |
| aflC-R | CTGAAACAGTAGGACGGGAGC | |
| aflP-F | CGATGTCTATCTTCTCCGATCTATTC | Toxin synthesis structure genes |
| aflP-R | TCTCAGTCTCCAGTCTATTATCTACC | |
| aflO-F | CTTTCGGCAGTGACCTAACC | |
| aflO-R | TCTTGAACTATAAGGCGACCA | |
| aflR-F | AAAGCACCCTGTCTTCCCTAAC | |
| aflR-R | GAAGAGGTGGGTCAGTGTTTGTAG | |
| aflS-F | GCTCAGACTGACCGCCGCTC | Toxin synthesis regulatory genes |
| aflS-R | GCTCAGACTGACCGCCGCTC | |
| 18S rRNA-F | CTGAAGACTAACTACTGCGAAAGC | |
| 18S rRNA-R | GAGCGGGTCATCATAGAAACAC | |
| β-tublin-F | TTGAGCCCTACAACGCCACT | RNA extraction quality testing |
| β-tublin-R | TGGTTCAGGTCACCGTAAGAGG | |
| actin-F | ACGGTGTCGTCACAAACTGG | |
| actin-R | CGGTTGGACTTAGGGTTGATAG | Internal reference gene |

by Trimmomatic (v0.36) (*Liao et al., 2014*). The clean reads were mapped to the reference genome of *A. flavus* by Burrows-Wheeler Alignment tool (BWA, v0.7.15) (*Li and Durbin, 2009*). MACS2 (v2.1.1) and Bedtools (v2.25.0) were used for peak calling and peak annotation, respectively. Differential binding peaks were identified by Fisher's test with q-value<0.05. HOMER (v3) was used to predict motif occurrence within peaks with default settings for a maximum motif length of 12 base pairs (*Hull et al., 2013*). Genes less than 2000 bp away were associated with the corresponding peak. GO and KEGG enrichment analyses of annotated genes were implemented in EasyGO (*Zhou and Su, 2007*) and KOBAS (v2.1.1) (*Xie et al., 2011*), with a corrected p-value cutoff of 0.05.

## qRT-PCR analysis

The fungal spores ($10^6$/mL) were cultured in PDB medium for 48 hr, and then mycelium was ground into powder with liquid nitrogen. Total RNA was prepared by TRIpure total RNA Extraction Reagent (Bestek, China) according to the protocol used by Zhang (*Mengjuan et al., 2021*). qRT-PCR was performed according to a previous study (*Hu et al., 2018*), and the primers were shown in *Table 3*.

## Oxidative stress assays

To evaluate the role of SntB in fungal resistance to oxidative stress, a series concentration (0, 0.12, 0.24, and 0.36 mM) of MSB were added to the medium. $10^6$ fungal spores for each strain were inoculated on the medium and cultured in dark at 37°C. The diameters of colonies were measured 3 days after inoculation and the inhibition rate was calculated as previously described (*Pan et al., 2023*). The AFB1 product was analyzed by TLC after the strains were cultured in YES medium in dark at 29°C for 7 days.

## ROS assay

With the instructions provided in the user's manual, the intracellular ROS production was measured using a ROS assay kit (S0033S, Beyotime Institute of Biotechnology, China). After harvest, the mycelia were incubated with 10 µM DCFH-DA and 50 g/mL Rosup for 30 min. With SpectraMax Imaging Cytometer (Molecular Devices, Sunnyvale, CA, USA) at emission wavelength of 525 nm and excitation wavelength of 488 nm, then fluorescence signals of intracellular ROS production were acquired.

## Statistical analysis

All data in this study were expressed as mean ± standard deviation. The statistical analysis was performed using the software GraphPad Prism8 (GraphPad Software, La Jolla, CA, USA). The difference was considered to be statistically significant when $p < 0.05$.

## Acknowledgements

We especially thank Prof. Shihua Wang, Jun Yuan, Xiuna Wang, Yu Wang, and Xinyi Nie for their support in instrument maintenance and reagent ordering. This work was funded by the grants of the National Natural Science Foundation of China (No. 32070140), the Nature Science Foundation of Fujian Province (No. 2021J02026), and the State Key Laboratory of Pathogen and Biosecurity (Academy of Military Medical Science) (SKLPBS2125).

## Additional information

### Funding

| Funder | Grant reference number | Author |
| --- | --- | --- |
| National Natural Science Foundation of China | 32070140 | Zhenhong Zhuang |
| Natural Science Foundation of Fujian Province | 2021J02026 | Zhenhong Zhuang |
| State Key Laboratory of Pathogen and Biosecurity | SKLPBS2125 | Zhenhong Zhuang |

The funders had no role in study design, data collection and interpretation, or the decision to submit the work for publication.

### Author contributions

Dandan Wu, Conceptualization, Investigation, Methodology; Chi Yang, Conceptualization, Data curation, Writing – original draft, Writing – review and editing; Yanfang Yao, Validation, Investigation, Methodology; Dongmei Ma, Data curation, Validation, Investigation; Hong Lin, Investigation, Visualization; Ling Hao, Data curation, Methodology; Wenwen Xin, Resources, Methodology; Kangfu Ye,

Yule Hu, Validation, Investigation; Minghui Sun, Investigation, Methodology; Yanling Yang, Supervision; Zhenhong Zhuang, Supervision, Writing – review and editing

**Author ORCIDs**
Chi Yang ⓘ https://orcid.org/0000-0002-8617-3820
Zhenhong Zhuang ⓘ https://orcid.org/0000-0003-0968-6507

Reviewer #1 (Public review): https://doi.org/10.7554/eLife.94743.5.sa1
Reviewer #2 (Public review): https://doi.org/10.7554/eLife.94743.5.sa2
Author response https://doi.org/10.7554/eLife.94743.5.sa3

---

## Additional files

**Supplementary files**
• Supplementary file 1. The data of RNA-seq and ChIP-seq analysis. (a) Sequencing data statistics in RNA-seq. (b) Alignment results of each sample in RNA-seq. (c) The information of differentially expressed genes (DEGs) in transcriptome data. (d) Gene ontology (GO) enriched in the DEGs in transcriptome data. (e) Kyoto encyclopedia of genes and genomes (KEGG) pathways enriched of the DEGs in transcriptome data. (f) Sequencing data statistics in chromatin immunoprecipitation sequencing (ChIP-seq). (g) Alignment results of each sample in ChIP-seq. (h) The information of up-regulated peak in ChIP-seq data. (i) Known motifs identified by HOMER motif enrichment analysis software. (j) GO enriched in the up-regulated genes in ChIP-seq data. (k) KEGG enriched in the up-regulated genes in ChIP-seq data. (l) GO enriched in the 238 common DEGs in RNA-seq and ChIP-seq data. (m) KEGG enriched in the 238 common DEGs in RNA-seq and ChIP-seq data. (n) The information of DEGs related to lipase activity in transcriptome data. (o) The information of DEGs related to oxidative response in transcriptome data.

• MDAR checklist

**Data availability**
All data needed to evaluate the conclusions are present in the paper and/or the Supporting Information. Raw data of the ChIP and RNA-seq were submitted to GSE247683.

The following dataset was generated:

| Author(s) | Year | Dataset title | Dataset URL | Database and Identifier |
|---|---|---|---|---|
| Yang C, Zhuang Z | 2023 | SntB triggers the antioxidant pathways to regulate development and aflatoxin biosynthesis in Aspergillus flavus | https://www.ncbi.nlm.nih.gov/geo/query/acc.cgi?&acc=GSE247683 | NCBI Gene Expression Omnibus, GSE247683 |

---

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
