## [Editor Report · eLife assessment]

In this **useful** study, the authors investigate the regulatory mechanisms related to toxin production and pathogenicity in Aspergillus flavus. Their observations indicate that the SntB protein regulates morphogenesis, aflatoxin biosynthesis, and the oxidative stress response. The data supporting the conclusions are **compelling** and contribute significantly the advancing the understanding of SntB function.

---

## [Referee Report · Reviewer #1 (Public review)]

The study identifies the epigenetic reader SntB as a crucial transcriptional regulator of growth, development, and secondary metabolite synthesis in Aspergillus flavus, although the precise molecular mechanisms remain elusive. Using homologous recombination, researchers constructed sntB gene deletion (ΔsntB), complementary (Com-sntB), and HA tag-fused sntB (sntB-HA) strains. Results indicated that deletion of the sntB gene impaired mycelial growth, conidial production, sclerotia formation, aflatoxin synthesis, and host colonization compared to the wild type (WT). The defects in the ΔsntB strain were reversible in the Com-sntB strain.

Further experiments involving ChIP-seq and RNA-seq analyses of sntB-HA and WT, as well as ΔsntB and WT strains, highlighted SntB's significant role in the oxidative stress response. Analysis of the catalase-encoding catC gene, which was upregulated in the ΔsntB strain, and a secretory lipase gene, which was downregulated, underpinned the functional disruptions observed. Under oxidative stress induced by menadione sodium bisulfite (MSB), the deletion of sntB reduced catC expression significantly. Additionally, deleting the catC gene curtailed mycelial growth, conidial production, and sclerotia formation, but elevated reactive oxygen species (ROS) levels and aflatoxin production. The ΔcatC strain also showed reduced susceptibility to MSB and decreased aflatoxin production compared to the WT.

This study outlines a pathway by which SntB regulates fungal morphogenesis, mycotoxin synthesis, and virulence through a sequence of H3K36me3 modification to peroxisomes and lipid hydrolysis, impacting fungal virulence and mycotoxin biosynthesis.

The authors have achieved the majority of their aims at the beginning of the study, finding target genes, which led to catC mediated regulation of development, growth and aflatoxin metabolism. Overall most parts of the study are solid and clear.

Comments on revision:

The authors have thoroughly addressed all the concerns I raised. The current manuscript is robust and effectively presents evidence supporting its claims. The overall quality of the manuscript has significantly improved.

---

## [Referee Report · Reviewer #2 (Public review)]

The authors fully addressed my concerns and made appropriate changes in the manuscript. The quality of the manuscript is now significantly improved.

---

## [Author Response]

The following is the authors’ response to the previous reviews.

Thank you for your careful reviews of our manuscript. This revision is mainly aimed at addressing some minor errors in the text, English writing, grammar, etc. The details are as follows:

(1) We added the information for the *sntB*-HA strain in table 1.

(2) We added the primer information for the construction of *sntB*-HA strain in table 2.

(3) Some errors in English writing, grammar. Please see the revised manuscript with markers.